# Efficient Suppression of Abdominal Aortic Aneurysm Expansion in Rats through Systemic Administration of Statin-Loaded Nanomedicine

**DOI:** 10.3390/ijms21228702

**Published:** 2020-11-18

**Authors:** Natsumi Fukuhara, Yuto Honda, Nao Ukita, Makoto Matsui, Yutaka Miura, Katsuyuki Hoshina

**Affiliations:** 1Division of Vascular Surgery, Department of Surgery, Graduate School of Medicine, The University of Tokyo, 7-3-1 Hongo, Bunkyo-ku, Tokyo 113-8655, Japan; FUKUHARAN-SUR@h.u-tokyo.ac.jp; 2Laboratory for Chemistry and Life Science, Institute of Innovative Research, Tokyo Institute of Technology, 4259 Nagatsuta-cho, Midori-ku, Yokohama, Kanagawa 226-8503, Japan; honda.y.aj@m.titech.ac.jp (Y.H.); matsui.m.ad@m.titech.ac.jp (M.M.); 3Department of Life Science and Technology, School of Life Science and Technology, Tokyo Institute of Technology, 4259 Nagatsuta-cho, Midori-ku, Yokohama, Kanagawa 226-8503, Japan; ukita.n.aa@m.titech.ac.jp

**Keywords:** abdominal aortic aneurysm, drug delivery, polymeric micelle, statin

## Abstract

Abdominal aortic aneurysm (AAA) is a life-threatening disease. However, no systemically injectable drug has been approved for AAA treatment due to low bioavailability. Polymeric micelles are nanomedicines that have the potential to improve therapeutic efficacy by selectively delivering drugs into disease sites, and research has mainly focused on cancer treatments. Here, we developed a statin-loaded polymeric micelle to treat AAAs in rat models. The micelle showed medicinal efficacy by preventing aortic aneurysm expansion in a dose-dependent manner. Furthermore, the micelle-injected group showed decreased macrophage infiltration and decreased matrix metalloproteinase-9 activity in cases of AAA.

## 1. Introduction

Abdominal aortic aneurysm (AAA) is a common and lethal disease with a possible risk of rupture. There are multiple factors of progression including advanced age, history of smoking, male gender, and family history [1]. Ideally, the least invasive treatment for preventing AAA expansion is drug therapy, and various drugs have been used in AAA experimental models [2,3,4]. Doxycycline was the first antibiotic drug used to treat human AAAs in expectation of its anti-inflammatory effects; however, its preventive effect on AAA expansion has yet to be demonstrated [5]. A randomized trial using propranolol for small AAAs also did not demonstrate an effect on the expansion rate [6]. Generally, small drugs such as systemically injectable medicines are diffused throughout the body immediately after administration. Thus, it is fundamentally essential to adjust the appropriate dosage and treatment schedule to provide drug efficacy without side effects. However, the gap in biodistribution and pharmacokinetics between experimental animal models and humans is an unresolved issue in translational medicine [7], and possible reasons for the above failure include the short administration period as well as the relatively lower dose administered to humans compared with animal models. Therefore, the development of therapeutic strategies that can both control pharmacokinetics and provide selective drug delivery is an important topic in the study of AAA treatment.

Polymeric micelles form due to the self-assembly of polymer–drug conjugates (amphiphilic polymers). These are well-known as systemically injectable nanomedicines, comprising a dense biocompatible outer shell and a drug-incorporated core. This outer shell prevents drug–protein adsorption as well as facilitates both prolonged blood circulation and the spatiotemporal control of drug delivery [8,9,10]. Thus, targeted nanomedicines could supply valuable approaches to satisfy unmet medical demands. Some preclinical and clinical studies have shown effective therapeutic outcomes using nanomedicines incorporating cisplatin, DACH-platin, and doxorubicin with reduced toxic side effects of the payloads [11,12,13]. Although these studies have mainly involved cancer therapies, there are few challenges in vascular diseases, especially toward AAA treatment. In fact, we previously used poly-shelled nanoparticles incorporating rapamycin in an AAA rat model and demonstrated local accumulation of the nanoparticles in AAAs and decreased macrophage infiltration and matrix metalloproteinase (MMP) 2. This drug delivery system efficiently suppressed AAA expansion using a low dose of the drug [14].

Herein, we focused on statins, which are known to have pleiotropic effects besides lowering cholesterol levels and influencing pathophysiological mechanisms in atherosclerosis. The effect of statins toward AAA treatments remains controversial so far [15,16]. Some previous reports described that statins are currently one of the candidate drugs with favorable clinical data for reducing AAA growth, mortality, and rupture rates [17], but others claimed less impact for aneurysm growth in humans [18]. These trials were done using statin tablet, i.e., oral administration. Therefore, there is still room for investigating the effects and utility of drug delivery system by using statin-loaded nanomedicine to evaluate the formation and expansion of AAAs. Among statins, we selected pitavastatin, which has a strong anti-inflammatory effect on vascular walls [19]. We prepared a pitavastatin-introduced poly(ethylene glycol)-*b*-poly(l-lysine)-phenylboronic acid (PEG-PLys(FPBA). This polymer–drug conjugate was self-assembled in aqueous media to form a polymeric micelle. We studied the prevention of AAA expansion using an elastase-induced rat experimental AAA model to determine its potential as an injectable nanomedicine for vascular disease-targeted drug delivery.

## 2. Results and Discussion

### 2.1. Preparation and In Vitro Characteristics of Polymeric Micelles

Poly(ethylene glycol)-*b*-poly(l-lysine) (PEG-PLys) and PEG-PLys(FPBA) were synthesized according to our previous research [20,21], and their characteristics were measured by proton nuclear magnetic resonance (^1^H-NMR) and gel permeation chromatography (Appendix A). Briefly, PEG-PLys(FPBA) was dissolved into phosphate-buffered saline (PBS) and stirred with pitavastatin/tetrahydrofuran solution. The pitavastatin-loaded polymeric micelle (PS/m) was spontaneously formed upon ultrafiltration with water. In this process, new covalent bonds were formed between the phenylboronic acid on the polymer side-chains and the diol in statin, and then, those polymer–drug conjugates self-assembled to form polymeric micelles (Figure 1). The diameter of the polymeric micelles was approximately 50 nm with symmetric distribution (polydispersity index = 0.12), as determined by a zetasizer (Figure 2a). The micelle was determined to be relatively stable following 24 h incubation under physiological conditions (Figure 2b), indicating the potential for maintaining its micelle structure in the bloodstream.

We then prepared a fluorescent (Alexa 647)-labeled PS/m for confocal laser scanning microscopy (CLSM) analysis. After 24 h incubation, PS/m (red color) was internalized into the cytoplasm of smooth muscle cells (SMCs) and co-localized with lysosomes (green color), suggesting endocytosis-mediated cellular uptake (Figure 2c). In vitro cytotoxicity was also studied in SMCs. Comparing the 50% inhibitory concentrations (IC_50_) of PS/m and free pitavastatin, the IC_50_ value of PS/m was approximately 30 times lower than that of the free drug (Table 1). According to the CLSM result, the micelle was efficiently internalized by endocytosis and gradually released statin through pH-induced dissociation of covalent bonds between the phenylboronic acid and diol [22,23]. In contrast, pitavastatin is a dimer molecule (*F*_w_ = 880.98 g/mol), connecting two statins with Ca by a coordination bond, suggesting difficulty in bypassing the cellular membrane according to Lipinski’s rule [24].

### 2.2. Therapeutic Efficacy of Pitavastatin-Loaded Polymeric Micelles for AAA

After elastase infusion to the aorta, the rats were treated with intravenous injections of PBS (control) and PS/ms (dose = 2 mg/kg, 5 mg/kg, or 10 mg/kg on pitavastatin bases) (Figure 3a). After 7 days, the diameter of the AAAs in the PBS treatment group were estimated as 6.41 ± 0.43 mm, and those of the PS/m-injected groups with 2 mg/kg, 5 mg/kg, and 10 mg/kg dosage were 4.78 ± 0.43 mm, 3.88 ± 0.43 mm, and 3.63 ± 0.66 mm, respectively. The sizes of AAAs treated with PS/ms were significantly smaller than those treated with PBS (*p* = 0.042, 0.002, and 0.006, respectively) (Figure 3c,d). Furthermore, dose-dependent therapeutic efficacy that enhanced the inhibitory effect on aneurysmal growth was confirmed in the PS/m groups. Because no significant difference was detected in the sizes of the abdominal aortas between the four experimental groups immediately after the elastase infusion (Figure 3b), we concluded that these differences in the therapeutic outcomes were induced by the polymeric micelles used in our study. Importantly, five out of eight (62.5%) rats in the 10 mg/kg PS/m-injected group died during the treatment period, and all rats of the other treatment groups survived. Therefore, the dosage of 10 mg/kg of PS/m was assumed to exceed the maximum tolerated dose in this schedule. On the other hand, the blood biochemistry results showed no significant adverse effects following the PS/m treatment (Table 2), and no major macroscopic abnormalities were noted in the heart, liver, spleen, kidney, and muscle (Appendix A). These data indicated that a dose of PS/m under 5 mg/kg did not show acute toxicity via systemic injection. Of note, rhabdomyolysis is a concerning side effect of statin treatment [25], but our histological analysis revealed no such evidence, at least with this treatment protocol.

### 2.3. Histology after Treatment with Pitavastatin-Loaded Polymeric Micelle

The sections of aortas were stained with hematoxylin and eosin (HE) and Elastica van Gieson (EVG) stains for histological evaluation. The HE stain showed low accumulation of inflammatory cells around the aortic wall of the rats treated with PS/ms (Figure 4b–d), whereas a significantly higher accumulation was observed in the rats treated with PBS (Figure 4a,e). Additionally, the EVG stain showed the structural preservation of the medial elastic laminae in the PS/ms groups (Figure 4g–i) but were remarkably destroyed in the group receiving the PBS treatment (Figure 4f). The AAA samples were also assessed by immunostaining of α-smooth muscle actin (αSMA) and CD68. The immunostaining of αSMA revealed that αSMA-positive cells in media of AAAs were preserved in the PS/m treatment groups (Figure 4k–m), whereas few αSMA-positive cells were detected in AAAs PBS treatment group (Figure 4j,n). CD68 staining showed macrophage accumulation mainly in the adventitia of all experimental groups. The number of macrophages and CD68-positive cells after injections of PS/ms were scarce compared with those after injections of PBS (Figure 4o–s). Aneurysmal formation occurs in multiple steps, such as SMC apoptosis, acute or chronic inflammation, and extracellular matrix degradation [26]. Macrophages have been reported to play an important role in AAA development, leading to the destruction of the aortic matrix [26]. In our study, we observed inflammation and degradation of elastin in the aortic wall of elastase infusion model rats, suggesting that our PS/m could inhibit the accumulation of inflammatory cells (i.e., macrophages) around the aortic wall.

### 2.4. Gelatinatic Activity

AAA segments were collected from the treated rats and analyzed using zymography and profiling arrays of protein expression. Gelatin zymography successfully displayed gelatinase activities at 64 and 92 kDa, which corresponded to MMP-2 and pro-MMP-9, respectively (Figure 5). MMP-9 activity was markedly suppressed in the group treated with 5 mg/kg PS/m compared with the PBS-treated group. In contrast, no visible differences were observed between MMP-2 activities after treatment with PBS and PS/m.

MMPs, including MMP-1, -2, -3, -9, and -12, have strong effects against aneurysms, and statins could provide an anti-inflammatory effect and MMP inhibition [27,28,29]. Notably, MMP-9 is the most abundant MMP in AAA [30]. It was reported that MMP-9-knockout prevented AAA formation in mice, and some drugs suppressed MMP-9 expression with the structural prevention of aortic wall elastin and medial SMCs [27,31]. Thus, our results indicated that MMP-9 suppression might be one of the key factors for preventing the formation and expansion of AAAs, and the nanomedicine as systemically injectable drug developed in this study can be potentially utilized for AAA treatment. However, some additional experiments are needed to understand full details of the contribution of nanomedicine to reduce AAAs. Therefore, further research on identification of factors regarding formation and expansion of AAAs, detailed pharmacological studies, such as pharmacokinetics and toxicities, would clarify the association of PS/m and the action mechanism, and these are now ongoing as part of the development of newly discovered drugs and will be reported elsewhere.

## 3. Materials and Methods

### 3.1. Chemicals and Reagents

ε-Trifluoro acetyl-l-lysine *N*-carboxy anhydride [Lys(TFA)-NCA] was purchased from Chuo Kaseihin Co., Inc. (Tokyo, Japan). α-Methoxy-ω-propylamine-poly(ethylene glycol) (MeO-PEG-NH2, *M*_W_: 10,000) was purchased from NOF corporation (Tokyo, Japan). Dimethyl sulfoxide (DMSO) was purchased from Nacalai Tesque Inc. (Tokyo, Japan) and distilled with CaH_2_ (FUJIFILM Wako Chemicals corporation, Tokyo, Japan) under reduced pressure before use. Three-Carboxyl-4-fluoro-phenylboronic acid (FPBA) was purchased from Combi-Blocks (San Diego, CA, USA). Pitavastatin calcium was purchased from FUJIFILM Wako Chemicals corporation (Tokyo, Japan). PBS, 5M hydrochloric acid (HCl), 5M sodium hydroxide solution (NaOH(aq)), sodium chloride (NaCl), and 4-(4,6-dimethoxy-1,3,5-triazin-2-yl)-4-methylmorpholinium chloride (DMT-MM) were purchased from Wako Pure Chemical Industries, Ltd. (Osaka, Japan). Sodium hydrogen carbonate (NaHCO_3_), and d-sorbitol were purchased from Tokyo Chemical Industry Co., Ltd. (Tokyo, Japan). One-Methyl-2-pyrrolidinone (NMP), lithium bromide (LiBr), and trypsin/EDTA were purchased from Sigma-Aldrich Co. (St. Louis, MO, USA). Diethyl ether and methanol were purchased from Kanto Chemical Co., Inc. (Tokyo, Japan). Four-(2-Hydroxyethyl)-1-piperazineethanesulfonic acid (HEPES) was purchased from Dojindo (Kumamoto, Japan). Alexa647-NHS was purchased from Thermo Fischer Scientific (Waltham, MA, USA). Smooth Muscle Cell Growth Medium 2 was purchased from Promo Cell (Heidelberg, Germany).

### 3.2. Synthesis of PEG-Poly(l-lysine) Block Copolymer (PEG-PLys)

PEG-*b*-poly(l-lysine) (PEG-PLys) was synthesized according to our previous paper [19]. Lys(TFA)-NCA was mixed with MeO-PEG-NH_2_ as an initiator in distilled DMSO and stirred at 25 °C for three days in argon atmosphere. The reaction mixture was added dropwise into excess amount of diethyl ether to obtain PEG-poly(ε-trifluoro acetyl-l-lysine) PEG-PLys(TFA) as a white powder. ^1^H NMR spectra were recorded using AVANCE III 400 instrument (Bruker, Billerica, MA, USA) with *d*-DMSO. Gel permeation chromatography (GPC) of the polymer was obtained using an HPLC system (JASCO, Tokyo, Japan) equipped with two TSKgel superAW3000 and superAW4000 (Tosoh Corporation, Tokyo, Japan) in NMP containing LiBr (50 mM) at a flow rate of 0.3 mL/min at 40 °C. The *M*_W_/*M*_n_ of copolymers were calculated to be 1.12 on the basis of PEG calibration. The TFA group of PEG-PLys(TFA) was deprotected in a mixture of methanol/5M NaOHaq solution [4:1 (*v*/*v*)] at *r*.*t*. After overnight reaction, the mixture was dialyzed against 5 mM HCl solution and deionized water, and then lyophilized to obtain PEG-PLys as white powder. ^1^H NMR spectra were measured using AVANCE III 400 instrument (Bruker, Billerica, MA) with D_2_O, and the degree of polymerization (DP) of Lys unit was estimated to be 45 by comparing the ratio of peak area in butylene protons (PLys side chain) and oxyethylene protons (PEG backbone). A narrow molecular weight distribution of the PEG-PLys was confirmed by GPC (column: Superdex Increase 200 10/300GL, eluent: 10 mM phosphate buffer (pH 7.4) with 500 mM NaCl, flow rate: 0.5 mL/min, detector: UV-vis (absorbance = 220 nm)).

### 3.3. Synthesis of Phenylboronic Acid Modified Polymers [PEG-PLys(FPBA)]

PEG-PLys(FPBA) was synthesized as previously reported [19]. The conjugation of FPBA into the PEG-PLys side-chains was conducted using DMT-MM as condensation reagents in methanol containing 50 mM sodium bicarbonate and 25 mg/mL d-sorbitol solution (pH 8.5). After overnight reaction, the reaction mixture was dialyzed against 5 mM NaOH, and deionized water, followed by lyophilization to obtain PEG-PLys(FPBA). The rate of FPBA modification on polymer was calculated to be 40% from the peak intensity ratio of the propylene protons in PLys side chain to the protons of FPBA group by ^1^H NMR in D_2_O containing d-sorbitol (90 mg/mL). GPC analysis of the obtained polymers was also performed (column: Superdex Increase 200 10/300GL, eluent: 10 mM phosphate buffer (pH 7.4) with NaCl (500 mM) and d-sorbitol (500 mM), flow rate: 0.75 mL/min, detector: UV-vis (absorbance = 220 nm)).

### 3.4. Alexa647 Conjugation to PEG-PLys(FPBA)

Alexa647-NHS (1 *eq* to the polymer) and PEG-PLys(FPBA) were mixed in 50 mM NaHCO_3_ (pH 8.5) overnight for labelling of the polymer. For pre-purification, the mixture was dialyzed against deionized water (MWCO: 6–8000). After lyophilization, the obtained product was dissolved in 1 M NaCl and 500 mM d-sorbitol aqueous solution and further purified using PD-10 column (Sephadex THdeionized water (MWCO: 6–8000) and lyophilization.

### 3.5. Preparation of Pitavastatin-Loaded Polymeric Micelle

Pitavastatin/tetrahydrofuran solution (130 mM, 3.0 mL) was dropwise to PEG-PLys(FPBA)/PBS solution (0.60 mM, 30 mL) and stirred for 10 min. The mixture solution was purified by ultrafiltration (MWCO: 10 kDa, Millipore, Burlington, MA, USA). To obtain uniform micelle, the solution of statin micelle was passed through a microfluidic mixer (NanoAssemblr, Precision Nanosystems Inc, Vancouver, BC, Canada). For preparing Alexa647 labelled micelle, statin solution (13 mM, 100 µL) was mixed with Alexa647 labelled PEG-PLys(FPBA) solution (0.06 mM, 900 µL) in PBS for 10 min. The Alexa647 labelled statin micelle was purified in the same manner as described above.

### 3.6. Cell Culture

SMC cell line was grown in Smooth Muscle Cell Growth Medium 2 supplemented 1% PS at 37˙°C in a humidified 5% CO_2_ atmosphere.

### 3.7. Cytotoxicity

Cytotoxicity was evaluated using the Cell Counting Kit 8 (Dojindo) assay in SMC cells. SMC cells were seeded in a collagen I-coated 96-well plate (AGC TECHNO GLASS, Shizuoka, Japan) at a density of 2 × 10^3^ cells/well for 48 h. Free-statin solution (pitavastatin: 10 mM) was prepared in DMSO as control. The prepared control and statin micelle were diluted in cell culture media at various concentration. After 48 h incubation, the cells in each well were washed with PBS and incubated with fresh medium (100 μL) containing Cell Counting Kit 8 solution (10 μL) for 1 h. The absorbance at a wavelength of 450 nm was measured using a microplate reader (iMark, BIO-RAD, Hercules, CA, USA). The cell viability was calculated by the manufacturer’s instructions.

### 3.8. Confocal Laser Scanning Microscopic Observation

SMC cells (5.0 × 10^4^ cells/dish) were seeded into a collagen I-coated 35 mm glass dish (AGC TECHNO GLASS) and incubated for 48 h. The cells were washed with PBS and incubated in 1 mL of the Alexa647 labelled micelle (pitavastatin: 0.125 mg/mL) for 24 h. After the cells were washed with PBS, LysoTracker Red (Thermo Fischer Scientific, 1 µM) was then added to the solution to stain late endosome/lysosome. After additional 30 min incubation, the cells were washed with PBS and incubated with 16.2 µM Hoechst 33,342 (Life Technologies, Carlsbad, CA, USA) solution to stain the nuclei for 3 min. The cells were washed with PBS and observed in fresh PBS using LSM710. Alexa647, LysoTracker Red, and Hoechst 33,342 were excited using laser light at 633, 561, and 405 nm, respectively.

### 3.9. Elastase Infusion Model of AAA in Rat

All animal experiments were performed in accordance with the Guidelines for the Care and Use of Laboratory Animals as stated by The University of Tokyo (Permit Number: M-P18-115, approval date: 14 March 2019). We operated adult male Sprague–Dawley rats (300–400 g) for the experiments. They fed a normal diet and were kept in air-conditioned (21 °C ± 1 °C) with a 12 h light–dark cycle. All rats underwent elastase infusion of the aorta to induce AAA as described by Anidjar and Dobrin [32,33]. Rats were anesthetized with isoflurane. Laparotomy was performed in midline, and infrarenal aorta was isolated with ligation of lumbar arteries. A PE-10 tube was inserted into the infrarenal aorta from the left femoral artery. The proximal and distal parts of dissected aorta were clamped. Through the PE-10 tube, 10 units/mL of porcine pancreatic elastase (Type I; Sigma-Aldrich, St. Louis, MO, USA) was continuously infused through the PE-10 tube over 0.9 h with a microinfusion pump. After infusion, aorta was declamped, and the PE-10 tube was removed.

### 3.10. Therapeutic Effect of Pitavastatin-Loaded Polymeric Micelle

Rats were randomized into four treatment groups. Before and after the elastase infusion, maximum diameter of aorta was measured with microcaliper. Immediately after and at 2, 4, and 6 days after the operation, intravenous injections of the following solution were performed via tail vein. (i) 1 mL/kg PBS alone (*n* = 7), (ii) 2 mL/kg PS/m at a concentration of 5 mg/mL (*n* = 7), (iii) 5 mg/kg PS/m at a concentration of 5 mg/mL (*n* = 7), and (iv) 10 mg/kg PS/m at a concentration of 5 mg/mL (*n* = 8). At 7 days after the operation, the induced AAA was exposed under anesthesia. Measurement of maximum diameter of each AAA was performed. After the measurement, all rats were sacrificed and subjected to perfusion fixation with 4% phosphate buffered paraformaldehyde (0.1 mol L PO4, pH 7.3) at 120 mmHg. The segment of AAA was excised and immersed in the same fixative for 1 h. The excised aorta was processed for paraffin embedding (Figure 3a).

### 3.11. Histological Analysis of the AAA

Transverse cross sections (4 μm) were stained by HE or EVG staining in the standard manner. Immuno-stained for αSMA or CD68 were performed using the streptavidin–biotin complex peroxidase method with Histofine SAB-PO kit (Nichirei, Tokyo, Japan). After hydrogen peroxide treatment, blocking, and incubation with primary antibodies against αSMA (1:1000; ab5694, Abcam, Cambridge, MA, USA) and CD68 (1:1000; ab31630, Abcam, Cambridge, MA, USA), the sections were treated with biotin-labeled secondary antibody and visualized. Each section was photographed at original magnification ×40. Quantitative analysis of (i) cell density of HE stains, (ii) αSMA, and (iii) CD68 positive cell density were done with 25 pictures of each staining. The data were analyzed using one-way analysis of variance (ANOVA), and * *p* < 0.05 was considered significant.

### 3.12. Gelatinase Activity

Another set of rats was assigned to two treatment groups, and elastase infusion was performed. According to the same protocol as described, the rats of each group received intravenous injections of PBS (*n* = 5) and PS/m 5 mg/kg (*n* = 5) immediately after and at 2, 4, and 6 days after the elastase infusion. At 7 days after, the segment of AAA was excised from each rats and homogenized in 200 μL passive lysis buffer (Promega, Madison, WI, USA). Gelatinase activity of the tissue lysates was determined using a gelatin zymography kit (AK47, Primary Cell Co, Sapporo, Japan) according to the manufacturer’s instruction. Five micrograms of protein of each lysate were separated on SDS-PAGE. SDS-PAGE gel was incubated in incubation buffer for 40 h at 37 °C. MMP marker containing MMP-2, pro MMP-2, and pro-MMP-9 were applied in separate lanes. The gel was scanned using a WSE-6100LuminoGraphI (ATTO, Tokyo, Japan) for zymography analysis, and the densitometric analysis was performed using Image J software.

### 3.13. Statistical Analysis

The values are expressed as the means ± standard deviation. Dunnett’s test was applied to determine the significance of the differences compared to the control group. The differences between two groups were analyzed by an unpaired Student’s *t*-test, and the differences among four groups were analyzed by one-way ANOVA. A *p* value < 0.05 was considered to be significant.

## 4. Conclusions

Our study presents valuable knowledge for the development of statin-loaded polymeric micelles as nanomedicine for AAA therapy in rat models. Compared with free pitavastatin, systemic injection of the micelle could provide efficient suppression of AAA expansion through macrophage elimination and MMP-9 inhibition, at least as suggested from the blood analysis, histological studies, and zymography. Because statins are well-known drug candidates for vascular diseases (e.g., AAA treatment), as evidenced by previous clinical studies, our strategies of statin–polymer conjugation, micelle formation, and drug delivery are applicable to a wide variety of diseases. Hence, this tactic should have universal application for the future development of pharmacotherapy against vascular-related diseases.

## Figures and Tables

**Figure 1 ijms-21-08702-f001:**
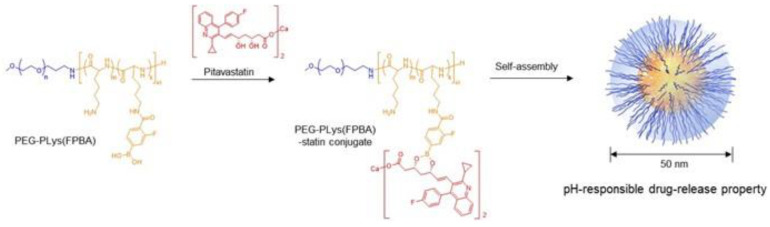
Schematic illustration of block-copolymer (blue = poly(ethylene glycol) segment, yellow = poly(l-lysine)-phenylboronic acid segment) and statin (red)-loaded polymeric micelles.

**Figure 2 ijms-21-08702-f002:**
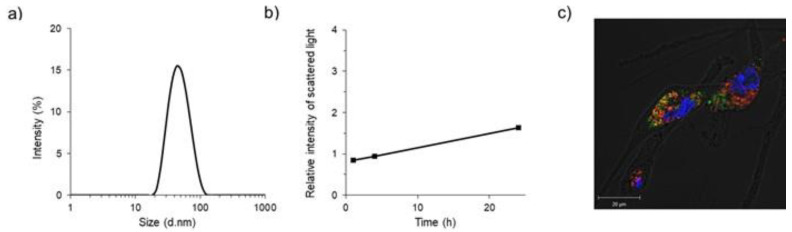
Characteristics of pitavastatin-loaded polymeric micelle (PS/m) and in vitro confocal laser scanning microscopy (CLSM) images. (**a**) Size distribution of PS/m in 10 mM phosphate buffer (pH 7.4), as measured by dynamic light scattering (DLS). Concentration: 0.5 mg-pitavastatin/mL, temperature: 25 °C. (**b**) Stability of PS/m in 10 mM phosphate buffer (pH 7.4) as measured by DLS. Concentration: 0.5 mg-polymer/mL, temperature: 37 °C. (**c**) In vitro CLSM of Alexa 647-labeled PS/m uptake in smooth muscle cells (SMCs). The cells were incubated with the micelle solution for 24 h at 37 °C. The picture shows merged images, which include the nuclei (blue), PS/m (red), and lysosome (green). Their colocalization is shown in yellow. Scale bars: 20 μm.

**Figure 3 ijms-21-08702-f003:**
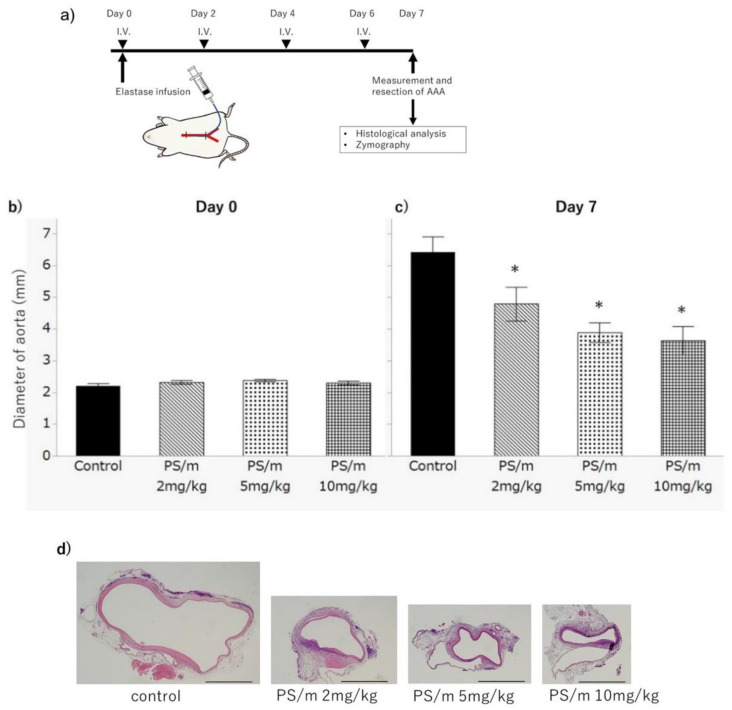
Experimental procedure and abdominal aortic aneurysm (AAA) treatments. (**a**) Experimental procedure of the elastase infusion model. Elastase infusion into the aorta is performed on day 0. Rats received an intravenous injection of the drug immediately after the operation and on days 2, 4, and 6. On day 7, the diameter of each aorta was measured, and it was resected for analysis. (**b**,**c**) Diameter of aortas on initial (day 0) and 7 days from the first administration. Control, PS/m 2 mg/kg, and PS/m 5 mg/kg groups: *n* = 7. PS/m 10 mg/kg group: *n* = 3. * *p* < 0.05. Error bars denote the standard error of the mean. (**d**) Cross sections of AAAs at 7 days, original magnification ×2. Scale bars: 1 mm.

**Figure 4 ijms-21-08702-f004:**
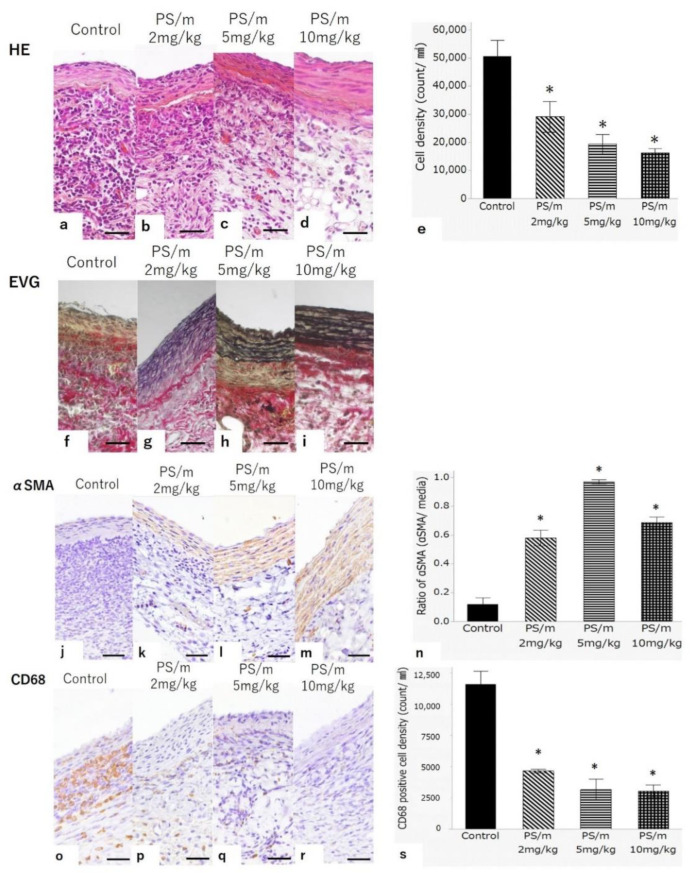
Histological appearance of AAA treated with phosphate-buffered saline (PBS) or PS/m. (**a**–**d**) hematoxylin and eosin (HE) stain, (**f**–**i**) Elastica van Gieson (EVG) stain, (**j**–**m**) α smooth muscle actin (αSMA), (**o**–**r**) CD68, original magnification ×40. Scale bar define 20 μm. Treated with PBS are (**a**,**f**,**j**,**o**); treated with PS/m 2 mg/kg are (**b**,**g**,**k**,**p**); 5 mg/kg are (**c**,**h**,**l**,**q**); 10 mg/kg are (**d**,**i**,**m**,**r**). (**e**) Cell density of HE stains. (**n**) Quantitative analysis of αSMA. (**s**) CD68 positive cell density. Elastic lamina (black in EVG) and αSMA positive cells (brown in αSMA) exist in media of AAAs, which was treated with PS/m groups (**g**–**i**,**k**–**m**). Abundant infiltrations of CD68 positive cells (brown) are observed in AAA, which was treated with PBS (**o**). * *p* < 0.05 compared with the control. Error bars denote the standard error of the mean.

**Figure 5 ijms-21-08702-f005:**
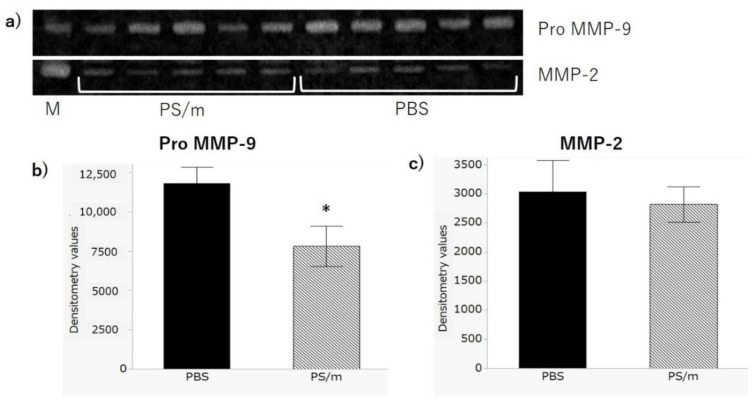
Gelatin zymography of AAA lysates. (**a**) Image of gelatin zymography. Upper bands (92 kDa) represent pro-matrix metalloproteinase (MMP)-9, and lower bunds (64 kDa) represent MMP-2. Samples are the AAAs of rats treated with PBS (*n* = 5) and PS/m 5 mg/kg (*n* = 5). MMP marker was applied in the left lane (M). (**b**) Gelatinase activity of pro-MMP-9. The activity of the group treated with PBS was significantly higher than the group treated with PS/m. (**c**) Gelatinase activity of MMP-2. There was no significant difference between the groups. * *p* < 0.05. Error bars denote the standard error of the mean.

**Table 1 ijms-21-08702-t001:** The 50% inhibitory concentrations with PS/m and free pitavastatin vs. smooth muscle cell.

	Free Pitavastatin	PS/m
IC_50_ (μM)	8.71	0.26

**Table 2 ijms-21-08702-t002:** Blood biochemical analysis on day 7.

	Control	PS/m 2 mg/kg	PS/m 5 mg/kg	PS/m 10 mg/kg
AST (IU/L)	62 ± 5.0	55 ± 4.2	61 ± 3.6	51 ± 7.9
ALT (IU/L)	21 ± 1.5	21 ± 1.3	21 ± 1.1	20 ± 2.4
BUN (mg/dL)	16.1 ± 0.9	17.0 ± 0.7	17.0 ± 0.6	18.3 ± 1.4
CRE (mg/dL)	0.25 ± 0.01	0.26 ± 0.01	0.26 ± 0.01	0.25 ± 0.02
CK (IU/L)	183 ± 31	210 ± 26	190 ± 22	154 ± 50
TG (mg/dL)	137 ± 11	72 ± 9 *	105 ± 8	56 ± 18 *
LDL-C (mg/dL)	9 ± 1.2	12 ± 1.0	12 ± 0.8	14 ± 1.8

Control, PS/m 2 mg/kg, and PS/m 5 mg/kg groups: *n* = 7. PS/m 10 mg/kg group: *n* = 3. AST, aspartate aminotransferase; ALT, alanine aminotransferase; BUN, blood urea nitrogen; Cre, creatinine; CK, creatine kinase; TG, triglyceride; LDL-C, low-density lipoprotein cholesterol. * *p* < 0.05 compared with the control.

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
