# Peer review of "Efficient Suppression of Abdominal Aortic Aneurysm Expansion in Rats through Systemic Administration of Statin-Loaded Nanomedicine"

_ijms, 2020, doi:10.3390/ijms21228702_

Round 1
Reviewer 1 Report
This is an interesting and well-written paper.
Author Response
Thank you very much for the comment. Please see the attachment.

Reviewer 2 Report
Over all the work is good but some improvement is needed towards introduction and results section . English needs to be improved.
Author Response
Thank you very much for the comment. We attached the certificate of the English editing. Please see the attachment.

Reviewer 3 Report
Efficient Suppression of Abdominal Aortic Aneurysm Expansion in Rats by Systemic Administration of Statin-Loaded Nanomedicine.
While the concept of using nanoparticles in AAA is interesting, it is not new and the authors have proven their strategy already with rapamycin. Nanoparticles are interesting to use the more dangerous or expensive drugs that could have serious side effects of would be too costly if given as oral drug.
My problem with this presented study with statins used as concept drug is that statins do not seem to impact aneurysm growth in humans, as is nicely demonstrated in a huge study of the Stanford AAA group. (Metformin prescription status and abdominal aortic aneurysm disease progression in the U.S. veteran Population by Nathan K. Itoga, et al; https://doi.org/10.1016/j.jvs.2018.06.194). In this study with >13000 AAA patients, 68% of them used statins and this was not correlated to a reduction in AAA growth, while metformin was. This is a secondary problem with statins, since most AAA patients are already on statins for cardiovascular problems/cholesterol homeostasis issues. Therefore statins are not an interesting class of drugs to implement in AAA patients, since they already use it. It seems that if the authors would put metformin in their nanoparticles there is novelty and thus potential interest, although metformin can also be used as regular drug since this has no side effects. The above is my main concern with the current manuscript.
Other issues:
Fig 3. An example of a cross section of an AAA for every condition would be appreciated.
Table 2. TG measurement, does that include all TG present in lipoproteins? Were these rats starved to obtain blood samples? Is it known which fraction, chylomicrons or VLDL is reduced upon statins? It seems that perhaps the statins also affect the liver. Could qPCR be performed on liver samples to provide data on key genes involved in lipid metabolism in liver that are/are not changed upon statin nanoparticles?
Fig 4 is not very informative and could be omitted or provided as supplemental data.
Fig 5. This is a nice figure, but it all should be quantified into 4 graphs and cannot be just an illustrative figure! Moreover, collagenous fibrosis (black arrow) is hard to see. I have no clue what I should be seeing there. I just see elastin fibers. And the αSMA positive cells (black triangle) there are so many more SMC than the arrow is pointing at that the arrows seem useless here. Just state SMC in red (or brown). Same as for CD68 cells. A better white balance would be nice to improve the quality of the pictures. Then the color comes out nicer, now yellow background masks the red/brown. Were all pictures taken with the same magnification??
Fig 6. The reduced MMP9 is merely a readout of a reduction in CD68 macrophages. This means that the data are consistent, but it is an overestimation that this is the key point why there is reduced AAA. There are a thousand other events which are not measured that also contribute to reduced AAA. It has been already shown many times that reducing inflammation in and inflammation-driven AAA model will reduce AAA formation. There should be more novelty.
Where I see some potential is the following: The SMC in de media do not show ASMA staining any longer, yet the cells are still present (and not all apoptotic) in the media and thus SMC have likely changed phenotype. This phenotype switching is now widely known and a hot topic (nicely demonstrated by the following and others: Atheroprotective roles of smooth muscle cell phenotypic modulation and the TCF21 disease gene as revealed by single-cell analysis by Wirka RC, et al, Nat Med. 2019 Aug;25(8):1280-1289. doi: 10.1038/s41591-019-0512-5). Perhaps the authors can stain and quantify some markers as described in the above paper to show that the SMC have made this switch? That would make the manuscript novel, that statins (in a direct or indirect way) improve SMC phenotype switching.
Round 2
Reviewer 3 Report
While I trust the results from this manuscript, I am still concerned about the novelty of this manuscript.
Minor: I wonder if a t-test can be used comparing these 4 groups everywhere. I think this should be attended to.
Author Response
Reviewer 3’s comment:
- i) I wonder if a t-test can be used comparing these 4 groups everywhere. I think this should be attended to.
According to Reviewer 3’s suggestion, we have used the analysis of variance (ANOVA) and Dunnett’s test for our static analysis. The main revisions are listed below:
Original description:
The sizes of AAAs treated with PS/ms were significantly smaller than those treated with PBS (p = 0.016, 0.001, and 0.002, respectively) (Figure 3c and 3d).
Revised description (p3 line 118-120):
The sizes of AAAs treated with PS/ms were significantly smaller than those treated with PBS (p = 0.042, 0.002, and 0.006, respectively) (Figure 3c and 3d).
NEW Table 2:
Original description:
The data were analyzed using Student’s t-test, and *p < 0.05 was considered significant.
Revised description (p10 line 339-340):
The data were analyzed using one-way analysis of variance (ANOVA), and *p < 0.05 was considered significant.
Original description:
3.13. Statistical analysis
The values are expressed as the means ± standard deviation. Dunnett’s test was applied to determine the significance of the differences compared to the control group. The differences between two groups were analyzed by an unpaired Student’s t-test. A p value < 0.05 was considered to be significant.
Revised description (p10 line 354-358):
3.13. Statistical analysis
The values are expressed as the means ± standard deviation. Dunnett’s test was applied to determine the significance of the differences compared to the control group. The differences between two groups were analyzed by an unpaired Student’s t-test, and the differences among four groups were analyzed by one-way ANOVA. A p value < 0.05 was considered to be significant.
